

# Mutltifractality of Climate Networks

Adarsh Jojo Thomas[1], Jürgen Kurths[2], and Daniel Schertzer[1]

[1]Hydrology Meteorology & Complexity (HM&Co), École nationale des ponts et chaussées, IP Paris, 6-8 Av. Blaise Pascal, Champs-sur-Marne, France
[2]Department of Complexity Science, Potsdam Institute for Climate Impact Research, Telegrafenberg A31, 14473 Potsdam, Germany

**Correspondence:** Adarsh Jojo Thomas (adarsh-jojo.thomas@enpc.fr)

**Abstract.**

Geophysical fields are extremely variable over a wide range of space-time scales. More specifically, they are intermittent in the sense that the strongest fluctuations are increasingly concentrated on sparser and sparser fractions of the space-time domain. Multifractals have been developed to analyse and simulate across scales such multiscale intermittency, while climate
networks can detect and characterise extreme event synchronisation. In contrast to multifractal analysis, climate networks are usually generated at a given observation scale, despite displaying complex structures over larger scales and likely exhibiting similar complexity at smaller scales.

In this letter, we present how to overcome this dichotomy of approaches by analysing in detail the effects of increasing the observation scale for climate networks, as allowed by empirical data, i.e. how do they upscale. This must be understood as
a preliminary step to be able to downscale them, including for practical applications such as urban geosciences that require analysis and simulation of intermittent fields at very high resolution. This is one of the reasons why we are using precipitation to illustrate our multifractal climate network approach.

## 1  Introduction

Climate Networks (Tsonis et al., 2006; Donges et al., 2009; Zou et al., 2019) have been extensively used to study the long
range dependence/synchronization between different spatial locations of various geophysical fields using different statistical methods like cross-correlation (Tsonis et al., 2006; Donges et al., 2009), event synchronization (Malik et al., 2012; Boers et al., 2019), mutual information (Hlinka et al., 2013; Cellucci et al., 2005) to name a few. However geophysical processes are highly intermittent and varying in intensity which emerge from complex nonlinear interactions across different space-time scales. Networks are typically constructed at the available data resolution and do not account for possible inter/intra-scale interactions.
Recent advances in this direction have combined a wavelet based approach to tentatively account for synchronizations at different temporal scales (Kurths et al., 2019).

In contrast, multifractals (Schertzer and Lovejoy, 1987; Schertzer et al., 1997; Schertzer and Lovejoy, 1991; Lovejoy et al., 2010) provide a natural framework to analyze and simulate extremely varying and intermittent geophysical and environmental fields over various scales. The core idea is to regard these complex systems as a cascade of structures (Richardson, 1922; Ya-





glom, 1966; Mandelbrot, 1974) across a wide range of scales, thus generating long range nonlinear interactions between them. We here propose a substantial extension of the climate network approach to multiple scales in a systematic way, accounting for the intermittency and anisotropy in space-time highlighted by multifractals, which is able to capture dynamical behaviors of long range interactions.

## 2 Data and Methods

Although our multifractal climate network approach is quite general, we will introduce it and illustrate it with the Tropical Rainfall Measuring Mission (TRMM) satellite and gauge combined gridded precipitation dataset 3B42-v7 (Huffman et al., 2007, 2016). The TRMM dataset has a temporal resolution of 1 day and spatial resolution of $0.25° \times 0.25°$. The spatial span of the dataset is over the South Asian subcontinent (3.875°-38.875° N, 63.125°-97.125° E), having roughly 20,000 grid points, and for the total available time period of 22 years (1998-2019) restricted to the months from June to September (122 days)

typically associated with the monsoon activity.

### 2.1 Multifractal Analysis of Precipitation

Cascades and multifractals originate from turbulence and physically correspond to the concentration of the turbulent energy flux along the cascade to smaller and smaller fractions of the embedding space-time space. The large multiplicity of interactions can generate universal properties, which are defined by only a few relevant parameters that are further physically meaningful

(Schertzer and Lovejoy, 1987; Schertzer et al., 1997). The resulting Universal Multifractal (UM) framework has been often used to analyse the precipitation component of the weather and climate (Lovejoy and Schertzer, 2013). In order to analyse the rain rate $R$, the data is systematically coarse grained in space and/or time to decrease its resolution. Let $L$ denote the largest scale and $\ell$ denote the degraded resolution, then the scale-ratio $\lambda = L/\ell$ is the (dimensionless) resolution. The scale-ratios in space and time can be respectively denoted separately by $\lambda_s$ and $\lambda_t$ when both are degraded to different resolutions. They are

related by $\lambda_t \approx \lambda_s^{2/3}$ to account for the anisotropy in space-time (Schertzer and Lovejoy, 1987, 1989, 1991).

Let $\varphi_\lambda$ be the underlying cascading field which is conserved at all scales, i.e $\langle \varphi_\lambda \rangle =$ constant. Then the scaling parameter $H$ characterizes the deviation of $R_\lambda$ from conservation:

$$\Delta R_\lambda \approx \varphi_\lambda \lambda^{-H} \implies \langle |\Delta R_\lambda| \rangle \approx \lambda^{-H}. \tag{1}$$

where the sign $\approx$ means an asymptotic equivalence for large resolutions ($\lambda >> 1$), i.e. with prefactors $\neq 1$ and possibly coupled

with an equality in the probability distribution when random variables are involved. The normalized moments of $\varphi_\lambda$ scale with the moment scaling exponent $K(q)$ :

$$\frac{\langle \varphi_\lambda^q \rangle}{\langle \varphi_1 \rangle^q} = \lambda^{K(q)}. \tag{2}$$

Similarly the exceedance probability distribution of $\varphi_\lambda$ scales with the codimension function $c(\gamma)$ :

$$\Pr \left( \frac{\varphi_\lambda}{\langle \varphi_\lambda \rangle} \geq \lambda^\gamma \right) \approx \lambda^{-c(\gamma)}, \tag{3}$$





where $\gamma = \log(\varphi_\lambda / \langle \varphi_\lambda \rangle) / \log \lambda$ is the scale-invariant singularity. Both the moment and probability exponent functions are related by the following Legendre transforms (Parisi and Frisch, 1985):

$$K(q) = \max_\gamma (q\gamma - c(\gamma)), \quad c(\gamma) = \max_q (q\gamma - K(q)) \tag{4}$$

Under the UM framework, both $K(q)$ and $c(\gamma)$ of conservative multifractal fields depend only on the parameters $\alpha, C_1$ :

$$K(q) = \begin{cases} \frac{C_1}{\alpha-1}(q^\alpha - q), & \alpha \neq 1 \\ C_1 q \log(q), & \alpha = 1 \end{cases} \tag{5}$$

$$c(\gamma) = \begin{cases} C_1 \left( \frac{\gamma}{C_1 \alpha'} + \frac{1}{\alpha} \right)^{\alpha'}, & \alpha \neq 1 \quad (\text{with } 1/\alpha + 1/\alpha' = 1) \\ C_1 \exp\left( \frac{\gamma}{C_1} - 1 \right), & \alpha = 1, \end{cases} \tag{6}$$

where $0 \leq \alpha \leq 2$ measures the degree of multifractality, with $\alpha = 0$ for monofractal fields and $\alpha > 1$ for the class of multifractal processes having unbound extreme singularities, and $0 \leq C_1 \leq d$ is the codimension of the mean field ($d$ is the dimension of the embedding space). Both these parameters are directly estimated from Eq.2 using single trace moments (TM) (Schertzer and Lovejoy, 1987; Lavallée et al., 1992) from the conserved field at different scales in space and time to directly estimate the function $K(q)$. $H$ is estimated using the first order structure function (see Eq.1) and also from the spectral scaling exponent $\beta$ and the second order moment exponent $K(2) : \beta = 2H + 1 - K(2)$.

## 2.2 Precipitation Climate Networks using Time-delayed Mutual Information

Climate Networks are constructed between time series $R_i, R_j$ at different geographical locations called nodes or vertices $i, j \in V$ by connecting pairs of them with links (denoted by $i \sim j$) if a given measure of similarity $S(R_i, R_j) \geq 0$ of the pair is significant. Unweighted CNs can be constructed from the strongest similarities by thresholding the similarity matrix $S$ :

$$A_{i,j} = \Theta(S_{i,j} - \theta) - \delta_{i,j} = \begin{cases} 1, & S_{i,j} \geq \theta, \ i \neq j \\ 0, & \text{otherwise} \end{cases} \tag{7}$$

, where $A$ is used to denote the adjacency matrix of a network, $\theta$ is some threshold on $S$, $\delta(x)$ is the Kronecker delta to remove self links $i \sim i$ and $\Theta(x)$ is the Heaviside function. The links of a network can be directed, possibly implying a direction of causality between two connected time series. However, we will be focusing on undirected networks here and consequently to symmetric similarity and adjacency matrices. Network centrality measures like the degree ($\deg_i = \sum_j A_{i,j}$) , which gives the number of edges connecting a given node, is used to study a network's properties.

Time-delayed Mutual Information (TDMI) (Hlaváčková-Schindler et al., 2007; Hlinka et al., 2013; Haas et al., 2023; Cellucci et al., 2005) has been used here to estimate the general dependence between two time series. Let $R_\lambda(\boldsymbol{x}, t)$ denote precipitation time series at the position vector $\boldsymbol{x}$, time $t$ and resolution $\lambda$; then TDMI is calculated as follows:

$$I(R_\lambda(\boldsymbol{x},t), R_\lambda(\boldsymbol{y},t+\tau)) = \iint p(R_\lambda(\boldsymbol{x},t), R_\lambda(\boldsymbol{y},t+\tau)) \log \left( \frac{p(R_\lambda(\boldsymbol{x},t), R_\lambda(\boldsymbol{y},t+\tau))}{p(R_\lambda(\boldsymbol{x},t)) \, p(R_\lambda(\boldsymbol{y},t+\tau))} \right) dR_\lambda(\boldsymbol{x},t) \, dR_\lambda(\boldsymbol{y},t+\tau), \tag{8}$$







**Figure 1.** A set of graphs computing the UM parameters for time and space in the left and right columns respectively : Trace Moment (TM) analysis for small scales in time with scale-break at 16 days, and in space with scale-break at around 440 km. The moment scaling exponent $K(q)$ obtained from TM plots is used to calculate $\alpha, C_1$ in time and space, as shown in the second row. Power spectra is shown in the third row which was used to estimate $H$ from the spectral slope $\beta$.

where $p(R_\lambda(\boldsymbol{x}, t)), p(R_\lambda(\boldsymbol{x}, t) R_\lambda(\boldsymbol{y}, t + \tau))$ are the marginal and joint distributions respectively. TDMI can also be used by thresholding the rain to account for dependence between extreme rain events.





The TDMI measure has many interesting properties that are reviewed in the supplementary text. The TDMI based similarities between two time series $R_\lambda(\boldsymbol{x},t), R_\lambda(\boldsymbol{y},t+\tau)$ is then given by:

$$85 \quad S_\lambda(\boldsymbol{x},\boldsymbol{y}) = \frac{I_\lambda(\boldsymbol{x},\boldsymbol{y})}{\max\left(I_\lambda(\boldsymbol{x},\boldsymbol{x}), I_\lambda(\boldsymbol{y},\boldsymbol{y})\right)}, \text{ where } I_\lambda(\boldsymbol{x},\boldsymbol{y}) = \max_{\delta(\boldsymbol{x},\boldsymbol{y})<\tau\le\tau_\lambda} I\left(R_\lambda(\boldsymbol{x},t), R_\lambda(\boldsymbol{y},t+\tau)\right) \quad (9)$$

The delay at which the dependence is maximized between two time series is called the correlation delay $\tau(\boldsymbol{x},\boldsymbol{y})$ as a function of the position vectors. A maximal delay $\tau_\lambda$ is set, while computing the similarity between any two pairs of time series, to limit false link creation from unreasonably large $\tau(\boldsymbol{x},\boldsymbol{y})$. The denominator proposed above acts as a normalizing factor which accounts for the self information (auto-correlation) between time series.

## 3 Climate Networks at Multiple Scales

### 3.1 Coarse Field Networks (CFN)

As pointed out in the introduction, the change of scale for CNs can have at least two meanings:

i) It can correspond to coarse graining the rainfall data in space-time using the scale ratios from Sect. [2.1] and obtain networks at various larger scales, calculated here for up to 5 different scales in this letter (see Table 1).

ii) This can also be done the other way around, i.e., computing the climate network at a given scale and then degrading its resolution.

Actually, we select a range of scales where the UM parameters remain unchanged. These Multifractal Climate Networks (MCNs) are parameterised by $p_\lambda, \theta_\lambda$ and $\tau_\lambda$ which depend on the data resolution $\lambda$ and are defined below.

TDMI based CNs are constructed for two case scenarios: (A) Using similarity measure from the whole range of precipitation values, and (B) from extreme precipitation events above a certain threshold rainfall value, in fact a fixed singularity $\gamma$. Climate Networks are typically constructed for the case of extreme rain events. The threshold rain value can be calculated from the multifractal relation :

$$p_\lambda = \Pr\left(R_\lambda \ge \lambda^\gamma\right) \approx \lambda^{-c(\gamma)} \quad (10)$$

The corresponding parameter $p_\lambda$ was fixed at the largest resolution ($p_\Lambda = 0.15$ for case B) and calculated for the rest of the 105 scales using Eq.10 by keeping constant the singularity $\gamma$ (see Table 1 below for $p_\lambda$ values). However $p_\lambda$ values for case A are trivially equal to 1.

The adjacency matrix $A$ is obtained by thresholding the similarity matrix $S_\lambda$ with some threshold parameter $\theta_\lambda$ (see Eq.7). This $\theta_\lambda$ directly influences the network structure and its properties like degree, link distribution, clustering coefficient, etc. by controlling the density of links, denoted by $\rho_\lambda$. If $S_\lambda$ is assumed to be a multifractal measure, then a similar equation to Eq.10 110 is obtained relating $\rho_\lambda$ at different scales to the threshold singularity $\gamma_S$:

$$\rho_\lambda = \Pr\left(S_\lambda \ge \theta_\lambda\right) = \lambda^{-c_S(\gamma_S)}; \gamma_S = \log\theta_\lambda/\log\lambda \quad (11)$$



**Table 1.** MCN parameter values used for the computing CNs at 5 different scales are shown below. Case A $p_\lambda$ values are omitted from this table since it simply takes the complete range of rainfall at all scales.

| Time-scale (days) | Length-scale (km) | Case B : $p_\lambda$ (%) | Max lag $\tau_\lambda$ (time-step) | $\rho_\lambda$ (%) |
|---|---|---|---|---|
| 1 | 27.75 | 15 | 20 | 5 |
| 2 | 83.25 | 22 | 10 | 10 |
| 3 | 138.75 | 27 | 7 | 14 |
| 4 | 222 | 31 | 5 | 18 |
| 5 | 305.25 | 35 | 4 | 22 |

Finally, the $\tau_\lambda$ free parameter sets the maximal delay (lag) up to which the strength of dependence between two given time series is calculated. This maximal correlation time-scale is usually significantly larger than the average lifetime of rain events at a given resolution, which makes it difficult to estimate at different scales. CFN computations for $\tau_\Lambda > 40$ days were found to significantly change the degree structure, whereas the results remained consistent when varied between 6 and 20 days. Therefore, $\tau_\lambda = 20$ days was fixed for all resolutions considered here.

### 3.2 Scaled Climate Networks (SCN)

To analyse the effect of a change of scale on CNs and their properties (e.g., their degrees), we introduce a pair of operators:

– $U_\lambda$ for upscaling or coarse graining by a scale ratio $\lambda$ (not always explicitly stated, for the sake of simplicity)

– $C$ for constructing a climate network from a given data set

They have a more or less immediate meaning when applied to a given field $F$ and we have first considered the composed operator $C \cdot U$ applied to the TRMM rain field. We have also evoked 'the other way around', i.e. the composed operator $U \cdot C$, with the difficulty that $U$ needs to be generalised to be applied to a network instead of a field. In fact, $U$ should coarse grain both components of a network, i.e. nodes and links, while fields have only the former component and $U$ is then restricted (and defined) to only act on this unique component.

As an introductory example, we consider the $U$ upscaling operator for unweighted CNs. It seems straightforward to impose the following rather simple rule (Fig. 2 for illustration): two upscaled nodes $I$ and $J$ are linked (i.e. the upscaled adjacency matrix satisfies: $A_{I,J} = 1$) if and only if, they contain more than a given number $\theta_\lambda$ of linked pairs of nodes $\{i, j\}$ belonging respectively to either of the scaled nodes (e.g., $i \in I, j \in J$). A priori, for such Scaled Climate Networks (SCN), this threshold number $\theta_\lambda$ scales with the scale ratio $\lambda$ of the performed upscaling, similar to Eq.11.

Figure 3 plots the relative degrees (normalized to 0-1 range) computed for 5 different scales, by first constructing the network from field $F_\lambda$ at these different scales, to obtain $CU_\lambda(F_\Lambda)$ described in the previous section, and to proceed in the reverse order to obtain $U_\lambda C(F_\Lambda)$ respectively. The network operator $C$ acting on $F_\lambda$ has resolution dependent parameters and the $\lambda$ notation



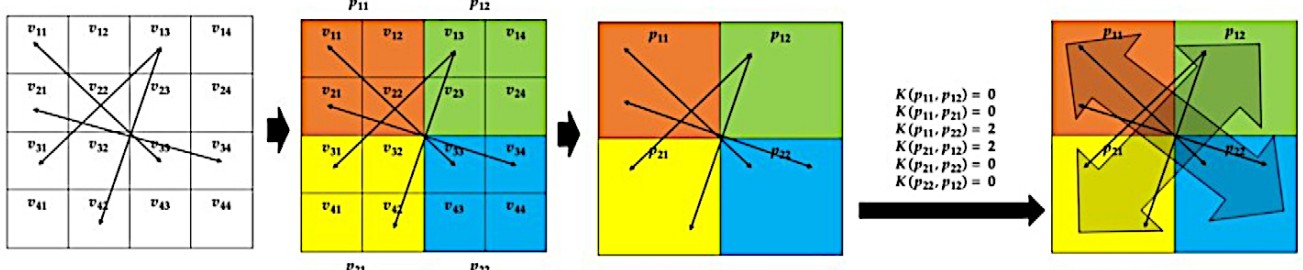

**Figure 2.** Scaled climate network schematic : Smaller nodes are grouped together, shown here in different colors, with links between nodes being replace by nodes between groups. The grouping of the node-set directly corresponds to the upscaling of the field in space.

has been reserved for the parameters alone (see previous section) for the sake of simplicity. Both these approaches are plotted for two cases (a) $p_\Lambda = 100\%$ and (b) $15\%$, with CFN and SCN plots in the top and bottom rows respectively. Moving from left to right, the hubs (high-degree nodes) over the region of North Pakistan diminish (reduced degree) at larger scales and simultaneously new hubs emerge in parts of central India. This further emphasises the need for a multi-scale approach to CNs for improved understanding of the climate dynamics at different scales. Differences in the network structures between these two approaches are very noticeable from both figures even for such a small range of scales. Local structural differences can be tentatively quantified with the Jaccard distance (Donnat and Holmes, 2018) defined below (where $|A|$ denotes the cardinality of the set $A$):

$$d_{\text{Jac}} = 1 - \frac{\sum([CU_\lambda F_\Lambda \cap U_\lambda CF_\Lambda])}{\sum([CU_\lambda F_\Lambda \cup U_\lambda CF_\Lambda])} = \frac{\sum|[C, U_\lambda]F_\Lambda|}{\sum([CU_\lambda F_\Lambda \cup U_\lambda CF_\Lambda])} \ , \tag{12}$$

which provides the relative number of common links between both ways of upscaling with respect to the total number of links, and is therefore a measure of their similarity. The second equality of Eq.12 shows that $d_{\text{Jac}}$ is also a metric of the relative non-commutativity of the operators $C$ and $U_\lambda$.

The Jaccard distance $d_{\text{Jac}}$ between CFN and SCN is plotted versus the logarithm of the resolution $\lambda_s$ in Fig. 4 for both cases A and B. It significantly increases with upscaling and corresponding resolution $\lambda_s$ decrease. The overall increase to $d_{\text{Jac}} \approx 60 - 70\%$ indicates that the local network properties have changed drastically for both cases A and B, although the increase for A is significantly lower ($\approx 10\%$) than for B, i.e. taking into account all the fluctuations, not only the extremes like for B. The scaled climate networks exhibit scale invariance in spatial distribution of the degree in contrast to CFN degree patterns which change with scale. This supports the limited commutativity of the network constructor $C$ with the upscaling operators $U_\lambda C \neq CU_\lambda$ for large upscaling, as already pointed out above.



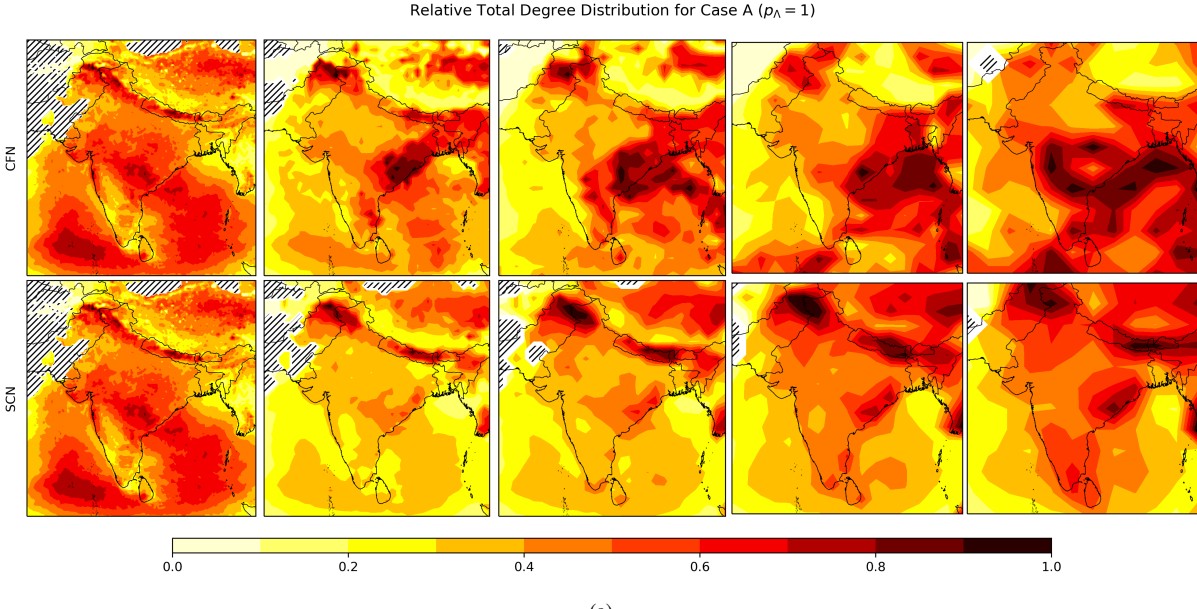

(a)

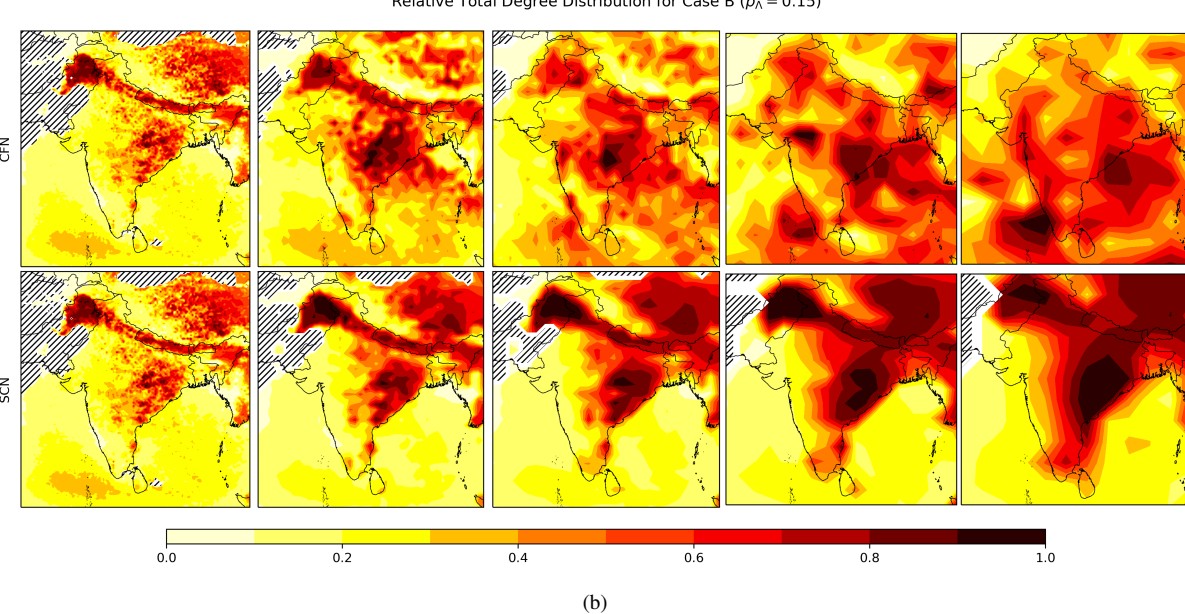

(b)

**Figure 3.** Relative Degrees for both approaches CFN (top row) and SCN (bottom row) at different scales $\lambda$ are plotted for two cases (a) Case A : $p_\lambda = 1, \forall \lambda$ and (b) Case B : $p_\Lambda = 0.15$ (see table 1 for all parameter values). The resolution in space decreases from left to right. The distribution of CFN's hubs (high-degree nodes) in space differ significantly with decreasing resolution compared to the SCN's hubs which are more stable and appear to be independent of it (and more prominent in (b)). The dashed regions were not included in the analysis for a lack of samples due to scanty rainfall (<15% rainy days during the period of study).



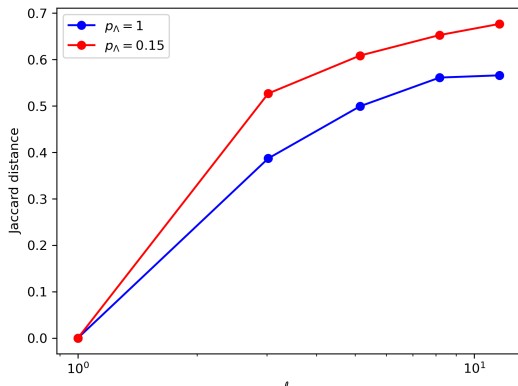

**Figure 4.** Scale by scale network dissimilarity between CFN and SCN displayed by the plots of the Jaccard distance $d_{\text{Jac}} \in [0, 1]$ vs. $\log_{10} \ell_s$ (cases A (blue) and B (red)).

## 4    Conclusions

This letter was devoted to analyzing the multifractality of climate networks, while their scale dependence is too often ignored.

This was achieved by first analyzing the space-time multifractal properties of a test field, which was also analyzed in the framework of climate networks. This enabled us to study the upscaling of climate networks. We first highlighted that there is not a single definition of an upscaling of climate networks due to a relative non-commutativity of two elementary operators, namely constructing a network from a given data set and coarse graining it. This was first shown on the degree of the networks, then we highlight the importance of the Jaccard distance as a theoretical and practical metric of the relative non-commutation.

The above results are fairly general, but the choice of monsoon rainfall data as a test field was motivated by many interests ranging from the social to the scientific. A common major interest is that of huge extremes, which have an overwhelming importance for understanding and modeling the water cycle. Scale dependent parameters were formulated, allowing for the systematic inference of network structures at larger scales. Our results not only confirm their scale dependence but also a significant sensitivity to the type of scale change, with the identification of new regions of interest emerging at larger scales in

the case of coarse-field networks previously undetected in single-scale network studies. We were able to capture the dynamics of processes with the potential inference of precipitation pathways dominating these scales. On the other hand, the scaled climate networks appear static, unable to identify any new regions.

To summarize, we showed the effectiveness and desirability to analyse climate data using a multiscaling approach. This approach could easily be extended to weighted sparse and dense networks. Also, the formalism developed so far is limited to

links between nodes at a given scale. Inter-scale links could be beneficial in further understanding cross-scale dynamics, for better simulations of climatic processes whilst preserving their spatial correlations and improved downscaling of networks and fields.



*Code availability.* The codes for generating the results are made by scripting Python software. All codes and data used in this study can be obtained from the corresponding author upon reasonable request.

*Sample availability.* The TRMM data can be accessed through https://doi.org/10.5067/TRMM/TMPA/DAY/7

*Author contributions.* AT carried out all the calculations, analysed the results and drafted the article. JK and DS co-designed and guided this study, facilitating access to the analysis and simulation software, respectively from PIK for climate networks and HM&Co for multifractals. All the authors contributed to the final version of the manuscript. DS provided funding for this project from École nationale des ponts et chaussées.

*Competing interests.* Some authors are members of the editorial board of journal Nonlinear Processes in Geophysics.

*Acknowledgements.* The authors acknowledge financial and computational support received from École nationale des ponts et chaussées.



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
