# Peer review of "Mutltifractality of Climate Networks"

_EGUsphere, 2024_

## Author Response (AR1)

**Point-by-point response document**

**#Referee1 Comment:**

Climate networks were introduced 20 years ago and ever since have provided a unique way to investigate dynamics, synchronization, and other properties in climate. As the authors state, climate networks are commonly generated at a given observation scale. Here the authors extend this by considering generation of networks at varying time scales akin to multifractals. The authors are leaders in the areas of climate networks and multifractals and I found their idea and approach ingenious. They show that analyzing networks using a multiscaling approach could lead to new understandings in climate dynamics.

Just a little minor point. The first time climate networks were introduced in the literature was in 2004 ("The architecture of climate networks", Tsonis and Roebber, Physica A, 333, 497-504). The Tsonis et al., 2006 was an expansion of the above work.

I highly recommend publication of the paper as is.

Anastasios Tsonis

**#Response:**

Thank you very much for your positive and encouraging feedback on our manuscript. We greatly appreciate your kind words and the recognition of our work.

We would like to make a small clarification that while our approach involves varying scales, the scales considered in this study pertain to both space and time, rather than time alone.

Thank you also for pointing out the original reference to the introduction of climate networks in 2004. We will ensure that the citation is adjusted accordingly.

We are grateful for your recommendation to publish and your valuable comments.

**#Changes:**

Citation Tsonis et al., 2006 was replaced by Tsonis and Roebber, 2004 on page 1, ll.14,16 and references modified on page 10, ll.240-241).

**#Referee2 Comments:**

The authors describe a new formalism to study the multi-scale/multi-fractal properties of functional network representations of climatological fields for the example of precipitation data.

Their reported intention to bridge the gap between two rather distinct methodological approaches is being addressed by thoroughly studying the network structures emerging at different spatial and temporal aggregation levels of the underlying observational data. I find the presented work very interesting and inspiring and generally worth being published in Nonlinear Processes in Geophysics. There are however a few aspects where I would recommend some clarification/revision of the present material prior to final publication.

**#Response:**

Thank you for your detailed analysis and positive overview of our work, as well as your valuable suggestions for improvement. We are pleased to read that you find our work interesting, inspiring and worthy of publication. Please find our detailed responses to your comments and suggestions below.

**#Changes:**

Please find the changes to the rest of the points elaborated below. Page and line references correspond to the revised manuscript.

**#Referee2 Comment 1:**

Main comments:

1. I do not find Fig. 1 referenced anywhere in the text.

**#Response 1:**

Apologies for the missing reference, Fig. 1 will be cited on page 3, ll.64,65 after Trace Moments (TM) and Spectral analyses.

**#Changes 1:**

Added missing references for Fig. 1 on page 3, ll.66,70 and page 4, l.103.

**#Referee2 Comment 2:**

Please ensure that this content is properly embedded and explained. With respect to the top

panels, I may note a clear deviation from the suggested power-law (emphasized by the linear fits in log-log scale) for lambda<10. It would be worth mentioning and explaining the reasons for this behavior (i.e., possible reasons for the scale breaks at 16 days and 440 km, respectively).

**#Response 2:**

You are right to ask for some physical explanations. The break around 16 days corresponds to the synoptic maximum in time. The scale break around 440 km is in agreement with estimates of the spatial scale for synoptic ISM (Indian Summer Monsoon) activity (see Malik et al., 2012), and disappears when samples from the whole year are analysed.

**#Changes 2:**

Physical explanations added to Fig. 1 captions.

**#Referee2 Comment 3:**

2. I am a bit concerned about practical issues in the estimation of the marginal and joint probabilities in the mutual information for precipitation data. As I learn from the Supplementary Material (which should be referenced in the text somewhere below Eq. (9)),

**#Response 3:**

We kindly remind the referee that the supplementary material has already been cited on page 5, l. 83, right above Eq. (9).

**#Changes 3:**

No changes.

**#Referee2 Comment 4:**

the authors use equi-quantile binning, which however seems incompatible with precipitation data, at least at their original (daily) temporal resolution. One should keep in mind that precipitation is, in stochastic terms, a combination of two (non-independent) stochastic processes, one occurrence process and one magnitude process. The amount of zero values in the data at a given grid point will likely not correspond well with the employed equi-quantile

binning, potentially causing a bias in the TDMI estimates.

**#Response 4:**

Let us clarify that to estimate local probability densities, the equi-quantile binning approach applies to all quantiles with the only exception for the lowest quantile, where it estimates probabilities taking into account the discrete nature of the zeros. Moreover, the denominator of the similarity in Eq. (9) already accounts for zeros and normalizes the TDMI. Therefore, the numerous zeros do not introduce systematic bias in CFN computation.

This was verified numerically against other mutual information estimators based on kNN (k-nearest neighbors distances, see Kraskov et al., 2004) and kernel density (see Moon et al., 1995), which produced similar TDMI results and CFNs, the results for which will be included in the supplementary.

**#Changes 4:**

New section with results showing the similarities between different TDMI estimators was added in the supplementary.

**#Referee2 Comment 5:**

Page 6, l.123: Could the framework of node-splitting invariant network measures (Heitzig et al., Eur. Phys. J. B, 85, 38, 2012, also co-authored by one of the authors) provide a more natural and theoretically meaningful way to aggregate/upscale (but also to disaggregate/downscale) spatiotemporal covariability information in the network domain?

**#Response 5:**

The node-splitting invariant approach allows upscaling only for nodes with identical neighbourhoods. Therefore it cannot guarantee CN upscaling at larger scales, as geographically adjacent nodes may have different neighbourhoods.

In contrast, the SCN approach is more general for CNs, since the requirement for an exact match of neighbourhoods is relaxed. Instead, it focuses on the overlapping neighbourhoods of geographically adjacent nodes. An 'upscaled' link is placed between two 'upscaled' nodes only when a significant number of links exist between the corresponding nodes at a given scale, capturing the strength of spatio-temporal dependence between two regions in space.

Both approaches are almost equivalent when the upscaling factor is small, although this will not be the case when one considers data near the poles.

**#Changes 5:**

No changes.

**#Referee2 Comment 6:**

4. Page 5, l.107: It is not clear if the TDMI threshold $\theta_\lambda$ is really considered scale-dependent or not, and if so, how exactly.

**#Response 6:**

It is scale dependent and follows a scaling law: $\theta_\lambda \approx \lambda^{\gamma_S}$. This relation is cited in Eq. (7).

**#Changes 6:**

Equation (11) is rearranged to better show the scaling law.

**#Referee2 Comment 7:**

This becomes potentially important along with page 6, l.128, where $\theta_\lambda$ is used, yet in a way that I do not quite understand. When $\theta_\lambda$ is the threshold of the similarity measure at scale $\lambda$ (which would be what I infer from Eq. (7) applied to individual scales), how can it be at the same time the number of links in the network at that scale? To me, this does not seem to make sense.

**#Response 7:**

The same symbol $\theta$ is used for the same action in SCNs and CFNs, i.e. thresholding upscaled similarities. However, the thresholding for SCN is indeed on the number of links between nodes, and we apologize for not being clear. However, both $\theta$ and $\rho$ are distinct, and represent thresholding of the similarity and density of the corresponding network, and should not be mistaken for one another. It could have been the source of confusion and this will be clarified in the revised version.

**#Changes 7:**

Note added on page 6, ll.137-139 for clarification of reuse of \theta_\lambda.

**#Referee2 Comment 8:**

Maybe there is just something in the explanation within the text that needs to be rectified; otherwise, I would cast quite some doubts on whether the reported results of the performed case study can really be interpreted in the way they are. (For example, if the absence of commutativity of the U_\lambda and C operators is partially due to improper/inconsistent thresholding at different scales.)

**#Response 8:**

We used the same \rho for both CFN and SCN to check for local differences between their networks. There is no inconsistency in the definition or usage of \theta and \rho, and this cannot be the reason for non-commutativity since neither of the two change the spatio-temporal dependencies in CFNs. This was also checked and confirmed for different \theta and \rho values.

**#Changes 8:**

No changes.

**#Referee2 Comment 9:**

Minor comments:

5. Page 1, l.14: Reference Zou et al. describes approaches for transforming individual time series into different classes of complex network representations, but not the generation of network structures from multivariate or spatio-temporal data. Citing it as a reference for climate networks thus appears inappropriate.

**#Response 9:**

Zou et. al. (2019) gives an introduction to multilayer and coupled networks, which could be used to study inter-scale network properties of the MCN. We apologize for embedding the citation here. It will be shifted appropriately or removed.

**#Changes 9:**

The citations and references for Zou et. al. (2019) have been removed.

**#Referee2 Comment 10:**

6. Page 1, ll.14/15: The term "long-range dependence" is commonly used with respect to temporal dependencies, while the spatial aspect commonly addressed in climate networks (or general functional network representations of spatially distributed time series) rather refers to what is called "teleconnections" in climate science. I do not think that referring in the present context to long-range dependence is well justified.

**#Response 10:**

We will clarify that we use the term 'long-range dependence' both in space and / or time, while 'teleconnection' is typically used for spatial long-range connections in climatology.

**#Changes 10:**

Added 'teleconnection' to the 'long-range dependence' terminology on page 1, l.15 indicating that in this context both terminologies are used to represent the same idea. However, later occurrences of 'long-range' refer to general interactions/dependencies in both space and time (for example on page 2, l.25).

**#Referee2 Comment 11:**

7. Page 1, l.17: Regarding the references in the previous line 16, all cited works refer to applications of different association measures to climate network analysis. This also applies to Hlinka et al. 2013 for the case of mutual information, but not to Cellucci et al 2005. I would recommend removing this citation at this point. Alternatively, you may cite the first-time application of mutual information to climate network studies by Donges et al., EPL, 87, 48007, 2009 (also co-authored by one of the authors).

**#Response 11:**

Cellucci et al introduces the equi-quantile binning approach, along with other binning methods, which makes it relevant. However, we will consider removing it since it has already been cited later on in the supplementary. The suggested reference of Donges et al 2009 has also already been cited above on ll.14,16.

**#Changes 11:**

Citations and references to Donges et al 2009 replace Cellucci et al 2005 on page 1, l.17, but no changes are made in the supplementary.

**#Referee2 Comment 12:**

8. Page 1, l.21: If I am not mistaken, the first combination of wavelet analysis and event synchronization was Agarwal et al., Nonlin. Proc. Geophys., 24, 599-611, 2017 (again also co-authored by one of the authors). Since I generally think that the number of self-citations by Prof. Schertzer and Kurths is already rather high, I would leave it to the authors whether or not to add yet another reference from their own impressive body of past works.

**#Response 12:**

Apologies for missing this citation which should have been in place of the current one. We will carefully consider whether to add or substitute it.

**#Changes 12:**

Citation and reference to Aggarwal et al 2017 added alongside Kurths et al 2019 on page 1, l.21.

**#Referee2 Comment 13:**

9. Page 2, Section 2: If I am not mistaken, the original TRMM dataset has both a higher temporal resolution than 1 day, larger spatial domain (circumglobal sub/tropics) and complete annual coverage beyond the JJAS season (likely chosen here because it is the Indian summer monsoon (ISM) season). It might be worth clarifying that the present study uses just a part of the TRMM data for a given region, with daily aggregates of the precipitation product (likely to avoid issues with the pronounced yet space-dependent diurnal cycle) and a focus on the ISM.

**#Response 13:**

Thank you for your suggestion, we should have clarified that only a subset of the spatial and temporal domain of the TRMM dataset was used to study the ISM. Daily aggregates of the available 3-hourly high resolution data also greatly reduces the size of the data and simplifies

the handling of such large datasets.

**#Changes 13:**

Changes made to page 2, Sect. 2 'Data and Methods', ll.32-37, clarifying the use of a subset of the full dataset and daily aggregates with brief explanation.

**#Referee2 Comment 14:**

Besides a possible interest to tie in with previous studies by Malik et al., Stolbova et al. and others, another well acceptable reason for focusing on the Indian subcontinent and the summer monsoon is to keep the number of nodes at the highest spatial resolution in a range where computing network properties is still computationally feasible. In general, I would very much welcome some more elaborations of the motivations of the authors for choosing the described setting.

**#Response 14:**

Thank you for your remarks and questions. Indeed, the primary goal of our paper is methodological and presents the MCN formalism within the context of ISM. The region is large enough to include some range of teleconnections while maintaining the computational feasibility previously mentioned, and is backed by extensive literature.

**#Changes 14:**

Changes made to the same lines as explained above.

**#Referee2 Comment 15:**

10. Page 2, l.45: Could you please clarify if the mentioned relation between the temporal and spatial scale ratios applies to 2d data, 3d data, or both?

**#Response 15:**

The scaling anisotropy relation between horizontal space and time is typically applied to 2D+1 space-time fields.

**#Changes 15:**

Changes made to page 2, ll.47-48.

**#Referee2 Comment 16:**

11. Page 2, l.47: Please mention that R_\lambda refers to the rain rate at resolution lambda.

**#Response 16:**

We thank the referee for pointing out this missing detail. We will make sure to explicitly mention R_\lambda as the rain rate at resolution \lambda.

**#Changes 16:**

Explicitly mentioned on page 2, l.50.

**#Referee2 Comment 17:**

12. Page 5, ll.93-96: I do not quite get the difference between the two cases mentioned here. At least, for me "degrading" would have the same meaning as "coarse-graining" or "upscaling".

**#Response 17:**

Indeed both the terms "degrading" and "coarse-graining" have exactly the same meaning as "upscaling".

**#Changes 17:**

No changes.

**#Referee2 Comment 18:**

Possibly the authors might have had something different in their mind; in this case, I recommend to elaborate a bit more explicitly on these two cases and their difference.

**#Response 18:**

On page 5 (ll.93-96), we introduce two approaches to MCN: CFN and SCN. Both are evaluated for two cases of the $p\_\lambda$ parameter. In case A, similarities across all rainfall values are used to construct CFN after upscaling the field, whereas in case B, only extreme values are considered for CFN similarities, also requiring field upscaling. The corresponding SCNs are constructed by upscaling the network for each case.

**#Changes 18:**

Changes made to perfectly clarify the differences between the two key approaches and the two cases on page 4, ll.101-102,105 , page 5, ll.109,128-129 and page 6, ll.141-144.

**#Referee2 Comment 19:**

13. Page 5, ll.100-101: "Climate Networks are typically constructed for the case of extreme rain events." – I rather disagree with this statement, since there are various other applications not even mentioned in this paper. Even if we restrict ourselves to precipitation networks, it rather depends on the temporal scale considered if the focus is on extremes or on general co-variability. Even for daily data, there has been work on correlation based precipitation networks (e.g. Ciemer et al., Climate Dynamics, 51, 371-382, 2018; again co-authored by one of the authors, or Ekhtiari et al., Chaos, 29 063116, 2019).

**#Response 19:**

The wide-range applicability of CNs has already been mentioned in the introduction. Currently, we treated two somewhat opposite cases: case A with the full range of rainfall intensities, whereas case B deals with only extremes (above a given threshold). It is an open point for future research to apply our approach to other climate variables and phenomena as well.

**#Changes 19:**

The problematic sentence was removed, and changes made on page 4, l.107 and page 5, ll.109-111 to clarify the specific use case of parameter $p\_\lambda$ in the context of the two cases.

**#Referee2 Comment 20:**

14. Table 1: I am wondering about the meaning of "length scale" in this table. Is it the mean distance between neighboring grid points? (Note that the grid points are on a regular latitude-longitude grid, where the spacing depends on geographical latitude).

**#Response 20:**

We assume each grid point is an identical square of side length 27.75 km at the smallest scale, which is the equivalent distance of 0.25° at the equator. This assumption is justified since the pixels at the boundary have a maximum deviation of 6 km.

**#Changes 20:**

Changes made to Table 1 title text.

**#Referee2 Comment 21:**

15. Page 6, l.115: What do you mean by "significantly change the degree structure"? Changes in the spatial pattern of node degrees?

**#Response 21:**

Yes, we mean it. Note that the spatial pattern of node degrees changed considerably for larger delays.

**#Changes 21:**

Changes made to page 5, l.120.

**#Referee2 Comment 22:**

16. Page 6, l.116: Do you really mean \tau_\lambda or rather \tau_\Lambda (with capital \Lambda)?

**#Response 22:**

Both \tau should be indexed by \Lambda. The \tau_\lambda parameter was fixed to 20 days for all \lambda. This corresponds with the max lag values mentioned in Table 1.

**#Changes 22:**

The correct symbol \Lambda is replaced in place of \lambda on page 5, l.121, and more text added explaining values for \tau_\lambda on page 5, ll.121-122.

**#Referee2 Comment 23:**

Technical comments:

    2, l.49: use Latex symbol \ll instead >>
    There are a few cases of references to equations in the main text where spaces are missing (ll.63, 65,105,107,109,130,144). In l.93, the Section number should not be given in brackets.
    3, ll.67,77: I suggest capitalization of "Delayed" in TDMI
    3, l.72: comma in the beginning of the line
    3, l.76: "are used"
    2, caption, l.2: "replaced"
    References: The individual citations should not contain publisher information in the case of journal articles, and also omit the term "number" (this also applies to the Supplementary Material)
    Reference Parisi & Frisch 1985) is missing information on the pages.
    12, l.223-224: eprint information appears redundant with the given URL/DOI.
    12, l.225: please capitalize the journal name

**#Response 23:**

We thank you for these technical comments that are all taken into account in the revised version.

**#Changes 23:**

All changes to the text and references are done.